# Chemoresistive Gas Sensors Based on Electrospun 1D Nanostructures: Synergizing Morphology and Performance Optimization

**DOI:** 10.3390/s24216797

**Published:** 2024-10-23

**Authors:** Aigerim Imash, Gaukhar Smagulova, Bayan Kaidar, Aruzhan Keneshbekova, Ramazan Kazhdanbekov, Leticia Fernandez Velasco, Zulkhair Mansurov

**Affiliations:** 1Institute of Combustion Problems, 172 Bogenbay Batyr Str., Almaty 050012, Kazakhstan; imash.aigerim@icp.kz (A.I.); kaidar.bayan@icp.kz (B.K.); a.keneshbekova@icp.kz (A.K.); zmansurov@kaznu.kz (Z.M.); 2Faculty of Chemistry and Chemical Technology, Al Farabi Kazakh National University, 71 al-Farabi Ave., Almaty 050040, Kazakhstan; kazhdanbekov_ramazan@live.kaznu.kz; 3International Chinese-Belorussian Scientiffc Laboratory on Vacuum Plasma Technology, Nanjing University of Science and Technology, 200 Xiaolingwei str., Nanjing 210094, China; 4Department of Chemistry, Royal Military Academy, Avenue de la Renaissance 30, 1000 Brussels, Belgium; leti_fv@hotmail.com

**Keywords:** chemoresistive gas sensors, electrospinning, 1D nanostructured materials, composites, electrospun fibers

## Abstract

Gas sensors are essential for safety and quality of life, with broad applications in industry, healthcare, and environmental monitoring. As urbanization and industrial activities intensify, the need for advanced air quality monitoring becomes critical, driving the demand for more sensitive, selective, and reliable sensors. Recent advances in nanotechnology, particularly 1D nanostructures like nanofibers and nanowires, have garnered significant interest due to their high surface area and improved charge transfer properties. Electrospinning stands out as a promising technique for fabricating these nanomaterials, enabling precise control over their morphology and leading to sensors with exceptional attributes, including high sensitivity, rapid response, and excellent stability in harsh conditions. This review examines the current research on chemoresistive gas sensors based on 1D nanostructures produced by electrospinning. It focuses on how the morphology and composition of these nanomaterials influence key sensor characteristics—sensitivity, selectivity, and stability. The review highlights recent advancements in sensors incorporating metal oxides, carbon nanomaterials, and conducting polymers, along with their modifications to enhance performance. It also explores the use of fiber-based composite materials for detecting oxidizing, reducing, and volatile organic compounds. These composites leverage the properties of various materials to achieve high sensitivity and selectivity, allowing for the detection of a wide range of gases in diverse conditions. The review further addresses challenges in scaling up production and suggests future research directions to overcome technological limitations and improve sensor performance for both industrial and domestic air quality monitoring applications.

## 1. Introduction

Gas sensors are indispensable in modern life, ensuring both safety and a higher quality of living [1]. As urbanization and industrialization accelerate, monitoring air quality has become more crucial than ever. In industrial settings, gas sensors help prevent accidents and detect toxic leaks, while in the automotive sector, they contribute to reducing harmful emissions [2,3]. In healthcare, these sensors facilitate early disease diagnosis through breath analysis [4], and in everyday life, they help maintain safer environments by continuously monitoring air quality. For gas sensors to be effective they must exhibit high sensitivity for detecting low gas concentrations [5], excellent selectivity to distinguish between various gases [6], and fast response and recovery times [7]. They also need to be stable and durable under different environmental conditions [8,9], consume minimal power, and be easily integrated into electronic systems [10].

The graph below illustrates the number of publications on gas sensors from 1980 to 2023, based on data from the Scopus database (see Figure 1). A sharp rise in publications since the early 21st century indicates a significant increase in scientific interest and research activity in this field. This surge is driven by the growing demand for more efficient and reliable sensors to meet challenges in industries such as manufacturing, healthcare, and environmental protection. Technological advancements, particularly the use of 1D nanostructures, have greatly enhanced the performance of gas sensors, which is reflected in the rising number of studies and publications in this area.

Gas sensors vary in their transduction mechanisms and include optical [11], electrochemical [12], electrical [13], and piezoelectric sensors [14]. Chemoresistive gas sensors belong to the electrical type of transduction, detecting and quantifying gaseous analytes by measuring changes in their electrical resistance. The primary working principle of chemoresistive sensors is based on the adsorption and desorption of gas molecules on the surface of the sensing material, which alters the charge carrier density and, consequently, the material’s electrical resistance [15]. The interaction of gas molecules with active centers on the surface changes the energy barrier for charge carrier movement, which forms the basis of the measurable signal.

Chemoresistive gas sensors can be classified according to various criteria, such as the type of sensing material, detection method, and application area. Based on the materials used, these sensors are divided into several categories: semiconductor metal oxide (SMOx)-based sensors [16], carbon-based materials [17], conductive polymers [18], and composites [19]. SMOx, such as SnO_2_, ZnO, and TiO_2_, are the most commonly used materials due to their high sensitivity and stability [20,21,22]. Carbon-based materials, like carbon nanotubes, graphene, and reduced graphene oxide (rGO), have gained attention for their unique electrical and mechanical properties [23]. Conductive polymers, such as polyaniline (PANI) and polypyrrole (PPy), are also used because of their flexibility and modifiability [24]. Composites, which combine different materials, offer enhanced sensor performance due to synergistic effects [25].

Various methods are employed to fabricate chemoresistive gas sensors, including chemical deposition [26], spray pyrolysis [27], sol-gel processing [28], and electrospinning (ES) [29,30]. Each method offers its own advantages and limitations depending on the desired sensor characteristics. One of the most versatile and cost-effective methods is electrospinning. This process involves applying a high voltage to a polymer solution or melt, resulting in the formation of a thin jet that elongates and solidifies into fibers as it moves toward a grounded collector [31]. The resulting nanofibers can be assembled into nonwoven mats or aligned arrays, depending on the collector configuration [32]. One of the key advantages of electrospinning is the ability to produce fibers with diameters ranging from tens of nanometers to several micrometers, enabling precise control over the material’s morphology to suit specific applications [33].

Advantages of 1D nanostructures, such as nanofibers and nanowires, include enhanced diffusion properties and efficient charge transport, which provide more active centers for gas adsorption compared to 2D and 3D nanostructures [34]. Their high aspect ratio, surface energy, and unique morphology contribute to improved sensor performance, including faster response times and lower detection limits [35].

Traditional gas sensors often face challenges such as low selectivity, slow response time, and instability in harsh environments [36]. Innovative sensors based on nanomaterials can address these issues, but they also encounter challenges such as the complexity of scaling production processes and material structure heterogeneity.

This review aims to provide a comprehensive analysis of current research on chemoresistive gas sensors utilizing 1D nanostructured materials fabricated by electrospinning. It will cover the fundamental principles and mechanisms of chemoresistive gas sensing, focusing on the influence of material morphology and composition on sensor properties; detail the electrospinning process and parameters affecting nanofiber structure and properties; review various materials used in electrospun gas sensors, such as metal oxides, carbon materials, conducting polymers, and composites, with examples from recent literature; examine post-annealing methods and modifications aimed at enhancing performance, emphasizing structural and chemical changes; offer a comparative analysis of different electrospun nanofiber gas sensors, including case studies from recent research, highlighting key factors influencing efficiency; and identify current challenges and propose future research directions to overcome existing limitations and improve sensor performance.

## 2. Basics of Chemoresistive Gas Sensors

Chemoresistive gas sensors can be categorized into n-type and p-type based on the primary charge carriers, which can be electrons or holes. They operate on the principle of gas interaction with these charge carriers in semiconductor materials, where the donor or acceptor properties of the gases alter the charge carrier concentration through adsorption and desorption processes on the sensor surface, as well as interactions with defects or active centers. The adsorption of oxidizing gases on n-type semiconductor surfaces reduces the concentration of charge carriers and increases resistance [37], whereas reducing gases release trapped electrons, thereby decreasing resistance [38].

### 2.1. Operating Principles of Gas Sensors

The primary mechanisms of sensor response include ionosorption of oxygen and redox reactions on the surface of materials such as SMOx [39,40,41]. During ionosorption, oxygen molecules adsorb onto the surface of SMOx, forming oxygen ions (O_2_^−^, O^−^, O^2−^) depending on the sensor’s operating temperature. For instance, at temperatures below 200 °C, O_2_^−^ predominates, whereas at temperatures above 250 °C, O^−^ and O^2−^ become more prevalent due to the dissociation of oxygen molecules and the capture of electrons from the conduction band [42]. These ionic forms of oxygen alter the electrical properties of the material, generating the sensor signal. Additionally, electrostatic interactions with ionic centers on the surface, such as metal cations, enhance oxygen adsorption and influence sensor sensitivity [43].

In n-type semiconductors, an electron-depleted surface layer (EDL) forms due to the extraction of electrons from the conduction band, while in p-type semiconductors, a hole-accumulation layer (HAL) develops due to the increased concentration of holes [44]. The polarity of the sensor response, which depends on the dominant charge carriers, significantly affects its sensitivity (see Figure 2).

In the presence of oxidizing gases, the resistance of n-type semiconductors increases due to the expansion of the EDL. Conversely, in the presence of reducing gases, the resistance decreases because of the contraction of the EDL and the return of electrons to the conduction band, which enhances conductivity and alters the bending of the energy bands at the grain boundaries of the material [45]. Oxygen adsorption and reactions with gases can also affect the height of the Schottky barrier, thereby altering the conductivity and, consequently, the resistance of the sensor [46]. It is important to note that oxygen ionosorption plays a crucial role in the sensor response mechanism, and this process can be enhanced by increasing the surface area and concentration of charged metal ions on the surface [47]. The magnitude of the resistive response is determined by the intensity of surface reactions and the number of active centers, such as oxygen vacancies, which facilitate gas adsorption and change the electrical properties of the material [48].

The surface of metal oxide materials (SMOx) used in gas sensors contains various acid-base centers that play a crucial role in the gas-detection process [49]. Lewis acidic centers, represented by coordinatively unsaturated metal cations, can accept electron pairs from gas molecules, while Lewis basic centers, consisting of lattice oxygen anions, donate electron pairs. Brønsted acidic centers, formed by bridging hydroxyl (OH) groups, can donate protons to gas molecules, whereas Brønsted basic centers accept protons [50]. The presence of terminal hydroxyl groups, resulting from the dissociative adsorption of water, also influences the sensitivity of metal oxides. Managing the concentration and nature of these centers allows for the optimization of sensor sensitivity and selectivity, thereby significantly enhancing their effectiveness in gas detection [51].

It is worth noting that different gases may interact with the acid-base centers through various mechanisms, and the nature of these interactions can be predicted using Pearson’s hard and soft acids and bases (HSAB) theory [52,53]. According to this theory, soft bases, such as reducing gases (e.g., hydrogen, ammonia), preferentially interact with soft acidic centers, while hard bases (e.g., oxidizing gases) interact with hard acidic centers [54]. This can serve as a useful tool in the design and optimization of gas sensor architecture to improve their performance characteristics.

### 2.2. Gas Sensor Performance Characteristics

Key characteristics that determine the effectiveness of gas sensors include sensitivity, selectivity, response time, and stability (see Table 1). These parameters are crucial for the development and application of sensors, ensuring their reliability and accuracy [55,56]. Sensitivity (indicated as R) is defined as the ratio of the sensor’s resistance in the presence of gas (R_gas_) to its resistance in clean air (R_air_). For a sensor to function effectively, this ratio should be greater than 1. Selectivity is expressed as the ratio of the sensor’s response to different gases. Response time is characterized by the period required for the sensor to reach 90% of the final value between R_air_ and R_gas_, while recovery time determines how quickly the sensor returns to 90% of this value. It is also important to note that the limit of detection is a significant parameter for gas sensors, as it defines the minimum detectable concentration of a gas [57,58]. Optimizing these factors ensures measurement accuracy.

Additionally, it is important to note that the performance of gas sensors is directly influenced by environmental conditions such as temperature and humidity, which can significantly affect gas adsorption processes on the sensor’s surface [78,79]. To minimize these impacts, various methods are employed, including the use of filters or temperature control systems [73,80]. Durability is also a critical factor, as many sensors are prone to degradation and aging due to exposure to harsh environments or contamination [81]. Utilizing materials with high resistance to external conditions, as well as protective coatings, helps extend the lifespan of sensors, ensuring their reliability and stable operation throughout their service life [82].

### 2.3. Materials Used in the Manufacture of Gas Sensors

The choice of sensing material is essential for optimizing the sensitivity, selectivity, and stability of gas sensors. Among the most widely utilized materials are ZnO, TiO_2_, SnO_2_, and WO_3_, each offering distinct advantages for different sensing applications [83,84]. They exhibit high sensitivity and stability in detecting various gases [85]. However, a limitation is their requirement to operate at high temperatures (>300 °C), which increases energy consumption [86]. To enhance the performance of oxide sensors, nanostructured materials such as nanowires [87], nanotubes [88], and nanosheets [89] have been developed, which increase surface area and improve gas diffusion [90]. For example, modifying ZnO with platinum or palladium (Pd) significantly enhances sensitivity due to the sensitizing effect and catalytic activity of the surface. On the other hand, conducting polymers like PPy and PANI are widely used for creating gas sensors. These materials are attractive due to their flexibility, low cost, and ability to operate at room temperature [91,92]. Polymers can be further enhanced by incorporating metals such as gold (Au) or platinum (Pt), which increase sensitivity through greater active surface area and catalytic activity [93]. Polymers can also be used in composites with carbon nanomaterials, improving gas-sensing properties. Transition metal dichalcogenides (TMDs), such as MoS_2_ and WS_2_, are of interest due to their two-dimensional structure and high specific surface area, making them ideal for detecting various gases [94]. These materials offer high sensitivity and stability, particularly for detecting toxic gases like NO_2_ and CO_2_ [95]. When combined with nanomaterials such as graphene, TMDs show enhanced performance due to synergistic effects. Recently, MXenes have garnered significant attention among two-dimensional materials, demonstrating high sensitivity to various gases and ease of functionalization to improve selectivity and stability [96,97]. Perovskites, such as LaCoO_3_ and SrTiO_3_, alter their electronic structure upon gas interaction, enhancing their sensitivity. For instance, the perovskite Cs_3_CuBr_5_ has shown high sensitivity to gases including hydrogen and ammonia [98]. Metal–organic frameworks (MOFs), such as ZIF-8 and MIL-101, are notable for their porous structure and large specific surface area [99,100,101]. These materials can be functionalized to enhance selectivity for target gases and exhibit high stability even in high humidity conditions [102].

### 2.4. One-Dimensional Nanostructures for Gas Sensors

One-dimensional materials such as nanotubes, nanofibers, and nanowires possess unique properties that distinguish them from traditional powders and films used in gas sensors [103,104,105]. Firstly, due to their linear geometry and high specific surface area, 1D nanomaterials provide enhanced charge transport, leading to increased sensor sensitivity [106]. This enables them to interact more effectively with target gases, resulting in a stronger response compared to 0D and 2D materials. Secondly, 1D structures can create shorter pathways for electron transport, which minimizes the sensor’s response time and improves its dynamic characteristics [107].

Recent research in the field of gas sensors has demonstrated the development of new materials and nanostructures that significantly improve sensor characteristics such as sensitivity, selectivity, and stability. For example, hollow SnO_2_ nanospheres have shown a significant increase in response to hydrogen, with response and recovery times of 56 and 216 s, respectively, at 100 ppm [108]. Meanwhile, thin SnO_2_ films have demonstrated a maximum response to ethanol at a concentration of 50 ppm, with response and recovery times of 259 and 214 s, respectively [109]. SnO_2_ nanofibers, synthesized through electrospinning, provided the fastest response and recovery—2 and 64 s to acetone at a concentration of 50 ppm [110]. This can be attributed to the structured pathway and orientation of electron transport in the fibers, whereas in 0D and 2D materials, this process can be disrupted due to the presence of multiple directional axes. These results highlight that materials of the same nature can yield different outcomes depending on their structure, deepening our understanding of electron transport processes (Figure 3a–c).

A comparison with other ZnO nanostructures, such as star-shaped and spherical particles, further confirms the advantages of nanofibers. For instance, star-shaped and spherical particles show response and recovery times of 25 and 150 s, respectively, to CO at a concentration of 200 ppm [113], while nanofibers demonstrate a rapid response (25–29 s) and recovery (12–17 s) to CO in the range of 1 to 20 ppm [114]. This indicates that during the interaction with nanofibers, the conductive channel facilitates rapid electron transport (Figure 3b). In the case of 3D structures, disordered morphology can obstruct electron transport due to the high potential at grain boundaries (Figure 3b,d), which can also occur when interacting with oxidizing gases. The interaction of oxygen, as an oxidizing gas, reduces the conductive channel, increasing the potential barrier for electrons and, consequently, raising the resistance. Unlike 0D and 2D structures, linear structures provide a more uniform distribution of the conductive channel, leading to more predictable changes in resistance (Figure 3).

Thus, the use of composite and hybrid materials, as well as the development of 1D nanostructures, opens new horizons for creating highly efficient gas sensors that can meet the requirements of a wide range of applications. These advancements underscore the need for further research in the field of nanomaterials to enhance sensor characteristics and expand their functionality.

## 3. Fundamentals of the Electrospinning Technique

ES is a key technology for synthesizing 1D nanomaterials, such as nanofibers, with diameters ranging from nanometers to micrometers [115,116]. This method allows for the creation of materials with unique properties by precisely controlling the morphology and structure of the fibers. By varying the composition of polymeric or composite solutions and adjusting process parameters (see Figure 4), such as applied voltage, solution feed rate, the distance between the needle and collector, and the type of collector used (static or rotating), one can control the size, porosity, alignment, and orientation of the fibers [117,118,119].

Furthermore, electrospinning enables the incorporation of various additives, including nanoparticles, catalytic agents, and other functional components, which opens opportunities for creating multifunctional materials with enhanced mechanical, electrical, and chemical properties [120].

The principle of the electrospinning process is based on the stretching of a polymer solution under the influence of an electrostatic field, during which solvent evaporation occurs, and fibers are deposited onto the collector. The stability of the Taylor cone, which determines the diameter and morphology of the fibers, is critical for producing high-quality nanofibers [121]. Instability of the cone can lead to non-uniformity or the formation of globular structures. Conversely, the cone’s height directly affects fiber diameter, allowing process optimization, as demonstrated in the work using polyvinylidene fluoride (PVDF) as the fiber-forming polymer under specific parameters: feed rate of 1 mL/h, voltage of 10.8 kV, and a distance of 10 cm [122,123].

In addition to cone stability, electrospinning is susceptible to various instabilities, such as axisymmetric Rayleigh instability, which results in bead-like fibers. In a recent study [124], the authors demonstrated that increasing voltage or polymer concentration helps reduce these instabilities (see Figure 5a–c). While slower solvent evaporation facilitates the formation of thinner fibers, residual charges can limit the mat’s thickness to 0.5–1 mm [125]. Viscosity, polymer molecular weight, and the concentration of the fiber-forming agent also critically influence nanofiber morphology, determining their diameter and structure [126,127]. Increasing viscosity and air pressure promotes the formation of thin, uniform fibers [128]. However, surface tension of the polymer solution plays an equally important role: high surface tension induces instabilities, whereas reducing it improves fiber uniformity. The use of solvents with low surface tension and the addition of surfactants further enhance fiber morphology [129,130]. Additionally, the solution’s pH significantly affects fiber size and morphology. At pH 2.6, thinner and more uniform fibers are formed compared to pH 8.5 (see Figure 5d–f) [131].

One study [133] demonstrated that the geometry of the collector also affects fiber morphology: a cylindrical collector provides better tension, while a disk-shaped one results in a higher Young’s modulus. The applied voltage plays a key role in the formation of the Taylor cone and the ejection of the polymer jet. For example, increasing the voltage enhances fiber stability and reduces deviations [134,135]. The collector’s rotation speed also influences fiber diameter. At low speeds (100 rpm), fibers are less stretched and form irregular structures, whereas increasing the speed to 500 rpm results in more uniform fiber distribution and reduced diameters (see Figure 5g–i) [132].

Controlling the flow rate of the polymer solution is critical for achieving the desired fiber morphology. While reducing the flow rate leads to thinner fibers (see Figure 6a–c) [136], increasing it can cause the formation of beads if the flow rate becomes too high [31,137]. The applied voltage during electrospinning significantly influences the diameter and morphology of the fibers. At low voltages, globular structures tend to form due to insufficient electric field strength to produce uniform fibers. The low charge at the needle tip and the collector results in jet instability and fiber structure variations. Doubling the voltage helps produce smoother and more uniform fibers (see Figure 6d–f) [138]. Additionally, the relative humidity of the environment has a significant impact on fiber morphology. Under high humidity (65%), fibers become porous and irregular, and their diameter increases. In low-humidity conditions (5%), fibers are smoother and defect-free, making their morphology more suitable for further applications (see Figure 6g–i) [139].

On the other hand, the distance between the needle and the collector also affects the diameter and morphology of the fibers. Increasing the distance promotes the formation of thinner fibers due to enhanced stretching forces. However, if the distance is too short, thicker fibers with bead-like structures may form. Optimizing this parameter is essential to obtain high-quality fibers [140].

The addition of nanomaterials such as fullerenes, carbon nanotubes, and graphene can significantly alter the morphology of the fibers, increasing their porosity and diameter, though this may reduce the specific surface area. However, post-annealing methods like sintering can improve the mechanical properties of the fibers without altering the base polymer [141].

For gas sensors to function effectively, the length and morphology of the nanofibers are crucial, as they influence the formation of conductive channels and the resistance response during gas interaction. Unlike 2D and 3D structures, 1D nanomaterials provide optimal charge transfer, minimizing losses. Fiber diameter, grain size, and crystallinity also play a key role: reducing the diameter increases the surface area, and decreasing grain size to the Debye radius maximizes the resistance change during gas interaction. The balance between graininess and crystallinity is critical for achieving optimal sensor sensitivity and stability [142].

## 4. Performance of Chemoresistive Gas Sensors Obtained by the ES Method

High sensitivity and short response time are key characteristics of gas sensors. Electrospun nanofibers, due to their high porosity and networked structure, enable efficient transport of analytes to the surface of the sensing material, significantly accelerating the sensor’s response time [143,144]. The advantage of the electrospinning method lies in its relative simplicity and cost-effectiveness, making the production of such nanofibers feasible for large-scale applications. Additionally, the recyclability of nanofibers contributes to sustainable resource use and reduces environmental impact [145]. To achieve optimal gas sensor performance, careful control over both the fiber morphology and the fabrication conditions is essential [111,146].

The process of making chemoresistive sensors based on electrospun nanofibers involves several key steps. The first step is to prepare the electrospinning solution, where the choice of components such as polymers, solvents, and sensing materials determine the final properties of the fibers, followed by the electrospinning process (see Figure 7).

A critical stage in the process is the thermal treatment (calcination), which removes the polymer matrix and converts the precursors into metal oxides. The calcination parameters, such as temperature, heating rate, and duration, directly affect the crystallinity and grain size, which in turn determine the gas-sensing properties of the resulting fibers [111]. There are several methods for forming electrodes based on electrospun nanofibers. One approach involves depositing the fibers onto integrated electrodes followed by calcination, which ensures strong adhesion between the components and enhances the sensor’s performance [147,148,149,150,151,152,153,154].

Another method involves the pre-coating of fibers onto ceramic tubular electrodes, which enhances adhesion and increases the number of active centers for interaction with the gas environment (see Figure 8). This approach improves the sensor’s sensitivity and simplifies the sensor assembly process [155,156,157,158].

In some studies, electrodes are formed directly during the electrospinning process. This method allows for optimized adhesion, precise control over the coating thickness, and improved electrode performance (see Figure 9) [159,160,161].

Therefore, the successful development of chemoresistive sensors based on electrospun nanofibers requires a comprehensive approach that includes material selection, optimization of electrospinning and thermal treatment parameters, and the development of effective methods for integrating fibers with electrodes. Careful management of these parameters is essential to achieve high sensitivity, stability, and selectivity in the sensors.

Moreover, achieving these characteristics necessitates consideration of the analyte’s nature and its interaction with the gas-sensitive material’s surface. The sensor’s effectiveness is influenced not only by the fiber morphology but also by the reducing and oxidizing properties of the analytes, which can affect the interaction mechanisms with the sensitive material. These factors directly impact the accuracy and reliability of measurements, underscoring the importance of a holistic approach in the development and optimization of gas-sensitive sensors.

### 4.1. Gas-Sensitive Characteristics to Reducing Gases

Detecting reducing gases such as acetone, hydrogen, and ammonia is critically important for safety, environmental protection, and health [162,163,164]. These gases are widely used in industry and agriculture, but their leaks can lead to toxic effects, explosions, and deteriorated air quality [165]. For example, hydrogen, while a promising fuel, is highly flammable if leaked, and ammonia is toxic at high concentrations [166,167]. Early detection of such gases in environmental monitoring and industrial safety allows for timely responses to potential threats, helps prevent accidents, and minimizes harmful impacts on the environment and human health [168].

One promising research direction is the use of composite nanofibers based on metal oxides. The shape and structure of these fibers significantly influence their gas-sensing properties, as demonstrated by numerous studies on the detection of reducing gases.

An effective approach to enhance sensor performance is the use of nanofibers doped with nanostructures. For example, research [169] has shown the effectiveness of Fe_2_O_3_ nanotubes doped with terbium (Tb) for acetone detection. These nanofibers, produced via electrospinning and calcined at 550 °C, exhibited a response of 53.2 at a concentration of 50 ppm acetone and 170 °C, which is 13 times greater than the sensitivity of pure Fe_2_O_3_ nanotubes. The doping with terbium further increases the concentration of oxygen vacancies, improving interaction with gas molecules. The unique morphology of the hollow-structured nanotubes explains the high sensor response (53.2 at 50 ppm acetone) and low detection limit (200 ppb).

The morphology of nanofibers is also critically important for the stability and sensitivity of composite-based sensors. For example, composite fibers of CuO/ZnO synthesized via electrospinning (see Figure 10a–c) exhibit a porous and scaly structure, which contributes to high sensitivity and stability towards hydrogen sulfide (H_2_S) [170].

Modifying the morphology of fibers with noble metals such as gold and palladium also significantly enhances their sensitivity. For example, In_2_O_3_ nanofibers doped with 2 at.% Au and 2 at.% Pd exhibit a marked improvement in response to CO due to the increased conductivity provided by the noble metals (see Figure 10d–f) [171]. This also highlights the importance of fiber morphology in ensuring sensor sensitivity, as it affects the interaction process between the gas and active surfaces.

Despite identical synthesis conditions, such as applied voltage and the distance between the needle and the collector during electrospinning, the fiber morphology can vary significantly. For instance, composite fibers of PANI/hollow Carbon#In_2_O_3_ exhibit a pronounced surface roughness and a hollow structure (see Figure 10g–i), which enhance the binding of ammonia molecules [172]. These fibers demonstrate stability at low gas concentrations (1 ppm), highlighting the crucial role of morphology control in ensuring both the stability and sensitivity of sensors.

Another important area is the development of nanocomposites based on rGO, which enhances the sensitivity and selectivity of sensors. In a study [173], NiO nanofibers wrapped with rGO were synthesized for ammonia detection. This nanocomposite is characterized by high conductivity and increased specific surface area, which improves sensor sensitivity and accelerates response time. NiO creates active centers for NH_3_ adsorption, increasing material conductivity through interaction with nickel vacancies. The rGO-NiO sensor demonstrated high sensitivity to NH_3_ at a concentration of 50 ppm, with a response time of 32 s and a recovery time of 38 s. The high repeatability of the response confirms the sensor’s stable performance over extended periods.

The use of carbon nanomaterials, produced via electrospinning, not only provides a high specific surface area but also imparts mechanical flexibility to the sensors, which is relevant for flexible electronics. In a study by Xing Fan and colleagues [174], a flexible sensor based on electrospun carbon nanofibers (CNF) decorated with ZnO nanoparticles (ZnO@CNF) was developed for ammonia detection. The electrospun carbon nanofibers, obtained through electrospinning, carbonization, and pre-oxidation, serve as a flexible substrate for the even distribution of ZnO nanoparticles, ensuring stable sensor performance under mechanical deformation. The formation of a p-n heterojunction between ZnO and CNF provides enhanced properties compared to the individual components. At room temperature, ammonia reacts with oxygen ions on the ZnO surface, leading to the release of electrons and a decrease in sensor resistance. The sensor operates effectively at 23 °C, unlike higher temperatures required for sensors based on pure ZnO. Under high humidity conditions, the sensor also responds to the presence of water, leading to the formation of hydronium ions (H_3_O⁺) and an increased response. The sensor exhibits short response and recovery times—5 and 18 s, respectively—and maintains its properties after multiple mechanical bends, making it reliable for use in flexible devices.

The use of nanowires and nanorods also enhances the gas-sensing properties of sensors by increasing the surface-to-volume ratio. In a study by Cai Z. and Park S. [175], SnO_2_ nanofibers doped with palladium and In_2_O_3_ were synthesized for hydrogen detection. Palladium nanoparticles facilitate the formation of Schottky barriers and catalytic activation, significantly improving hydrogen sensitivity. Doping with In_2_O_3_ resulted in a 24-fold increase in sensor sensitivity compared to materials containing only palladium.

Thus, employing nanoscale structures, such as nanotubes, nanofibers, nanorods, and nanocomposites, significantly expands the capabilities of gas sensors for detecting reducing gases. The application of these nanomaterials notably enhances key sensor parameters, including sensitivity, selectivity, and operating temperature. Furthermore, these sensors demonstrate high stability and durability, maintaining their performance under repeated mechanical deformations and prolonged use. This makes them particularly promising for applications in flexible electronics and environmental monitoring systems.

The table below provides a comparative analysis of sensors for detecting various reducing gases, detailing materials, electrospinning parameters, target gases, response and recovery times, operating temperatures, and detection limits, highlighting the diversity of approaches and their adaptation to various practical requirements. See Table 2.

Thus, electrospun nanofibers are promising materials for the detection of reducing gases due to their high sensitivity and stability.

### 4.2. Gas-Sensitive Characteristics to Oxidizing Gases

Detecting oxidizing gases such as nitrogen dioxide (NO_2_), ozone (O_3_), and sulfur dioxide (SO_2_) is crucial for mitigating environmental and industrial risks [187]. These gases can have serious health impacts, contribute to environmental pollution, and cause damage to industrial equipment. For instance, NO_2_ and SO_2_ are major pollutants that contribute to acid rain formation and deteriorate air quality in urban areas [188,189]. Ozone, on the other hand, poses health risks when present in high concentrations in the lower atmosphere. Reliable and sensitive sensors capable of detecting even trace concentrations of these gases are essential for environmental protection, emission monitoring, and preventing industrial accidents [190].

A promising research direction involves the use of composite nanofibers based on metal oxides [39]. The morphology of these fibers plays a crucial role in their gas-sensing characteristics, as demonstrated by numerous studies.

For example, in study [143], nanofibers made from a mixture of CuO and Fe_2_O_3_, produced via electrospinning, showed an enhanced response to NO_2_. Nanofibers with a 0.5CuO-0.5Fe_2_O_3_ ratio were particularly effective, forming a binary oxide CuFe_2_O_4_ that creates heterojunctions between oxides, thereby enhancing sensor sensitivity. The morphology of these fibers, developed during the electrospinning process, improves the contact between active surfaces and gas molecules, significantly affecting their gas-sensing properties. The spinning time also plays a crucial role in shaping the fiber structure, which directly impacts response characteristics.

Another example is study [191], which investigates porous NiFe_2_O_4_ nanofibers with nanocrystallites, showing high sensitivity to H_2_S and NO_2_. The porous structure of these fibers increases surface area and enhances interaction with gases, improving sensor response. In this case, electrospinning conditions such as voltage directly influence morphology and, consequently, sensor sensitivity to different gases.

Particular attention is given to sensors based on materials that utilize optical excitation through localized surface plasmon resonance. For instance, Au/SnO_2_ nanofibers synthesized via electrospinning exhibit enhanced gas sensitivity under UV irradiation due to the gold’s surface plasmon effect (see Figure 11a–c) [192]. The fine morphology of the fibers improves photon transfer and increases the number of active centers for gas molecule interaction.

It is also important to note that incorporating reduced graphene oxides into composite fibers improves the sensor recovery time, which is related to increased overall porosity and conductivity of the fibers. Specifically, rGO-PVDF/WO_3_ nanofibers demonstrate high sensitivity to SO_2_ due to their porous structure and enhanced surface area (see Figure 11d–f) [193]. See Table 3.

Thus, the morphology of nanofibers plays a crucial role in achieving high gas-sensing performance in sensors. The electrospinning method offers unique opportunities to tailor this morphology, allowing for customization of sensors for specific gas-detection tasks. Comparative analysis of various studies demonstrates that the fiber structure and electrospinning conditions significantly influence sensor response, sensitivity, and selectivity. This makes fiber morphology a critical factor in the development of advanced gas analyzers.

### 4.3. Gas-Sensitive Characteristics to Volatile Organic Compounds (VOCs)

The detection of volatile organic compounds plays a crucial role in air quality control and assessing their impact on human health and the environment [204]. VOCs, such as benzene, toluene, and xylene, are widely used in industry and consumer products, but their emissions into the atmosphere can lead to smog formation, greenhouse gas effects, and adverse health impacts, including respiratory diseases and cancer [205]. VOCs readily evaporate at room temperature, making them challenging to detect without highly sensitive sensors. Monitoring VOCs is increasingly relevant in the context of growing urbanization and rising industrial emissions. Modern gas sensors with high selectivity and low detection limits can detect even trace concentrations of VOCs, which is essential for risk assessment and compliance with environmental and health regulations [206,207].

Electrospinning is a method for producing ultrathin fibers with a high specific surface area and controlled structure, which is critical for achieving high sensitivity and selectivity in gas sensors.

Furthermore, increasing the number of oxygen vacancies can be achieved by using MOFs to create porous structures that retain the morphological properties of precursors after thermal treatment. This contributes to an increase in active oxygen content and improved gas-sensing properties. Recent studies have proposed using polycrystalline ZIF-8 to enhance adsorption characteristics, which allows for the creation of more oxygen vacancies and improved sensitivity to ammonia and formaldehyde. Nanofibers of Pt/ZnO-In_2_O_3_ derived from Pt/ZIF-8 provide a 2.7-fold greater response to formaldehyde compared to pure In_2_O_3_, with fast response and recovery times (see Figure 12a–c) [208].

The interaction of two-dimensional MXene materials with one-dimensional metal oxide semiconductors also contributes to improved sensor characteristics. Nanofibers of MoO_3_ and layered Ti_3_C_2_Tx MXene, produced using electrospinning and chemical etching, demonstrate outstanding results in detecting trimethylamine. The Ti_3_C_2_Tx MXene–MoO_3_ composite material shows high response and rapid response-recovery times due to its significant specific surface area, active centers, and p-n heterojunctions (see Figure 12d–f) [209].

Nanofibers of tin dioxide (SnO_2_) and cerium dioxide (CeO_2_) nanoparticles, produced using electrospinning and hydrothermal synthesis, show significant improvement in sensitivity for detecting liquefied petroleum gas. The SnO_2_/CeO_2_ composite sensor exhibits enhanced sensitivity and moisture resistance compared to pure SnO_2_, which can be attributed to an increased number of oxygen vacancies and the formation of a heterostructure (see Figure 12g–i) [210].

The addition of various components to the composite structure significantly influences the morphology, altering the fiber surface and their gas-sensing properties. For instance, a study produced tri-metallic composite fibers of ZnSnO_3_/ZnO, which, with their hollow and rough structure (see Figure 12j–l), show stability in ethanol detection [211].

Additionally, it is possible to create composite materials that are entirely metal-free. In one study, fibers made from polyacrylonitrile/polyaniline (PAN/PANI) via electrospinning exhibited outstanding performance in detecting trimethylamine, achieving detection limits below 6 ppb and providing excellent repeatability [212]. These results affirm the effectiveness of using nanofibers for creating highly sensitive and stable sensors for volatile organic compounds. The comparative analysis table of various sensors highlights their advantages and limitations, making the use of electrospun nanofibers a promising approach for enhancing the monitoring and control of volatile organic compound concentrations in various environments. These fibers possess unique properties, such as high specific surface area and a high density of active centers, making them particularly suitable for detecting low concentrations of various VOCs.

The table provides a comparative analysis of different sensors for detecting volatile organic compounds, including their key parameters and characteristics. It lists sensor materials, electrospinning parameters (flow rate, voltage, and distance between the needle and collector), target gases, response and recovery times, operating temperature, selectivity, detection range, and sensitivity concentration for each material. For instance, a Pt-SnO_2_-based sensor demonstrates high sensitivity to acetone with a detection range from 0.1 to 20 ppm and a response time of 13 s. Similarly, a Rh-SnO_2_-based sensor is also designed for acetone detection but features a broader detection range (90–200 ppm) and a slower response time (2 s). See Table 4.

Other sensors, such as MoO_3_-WO_3_ and Pd@Co_3_O_4_-ZnO, exhibit significant recovery times and wide detection ranges, making them suitable for various applications and types of VOCs. Comparing these sensors highlights their advantages and limitations to VOC monitoring. Therefore, the use of electrospun nanofibers for detecting volatile organic compounds represents a promising direction that can significantly enhance the monitoring and control of VOC concentrations in diverse environments.

## 5. Challenges and Future Directions

The increasing emissions of toxic gases are driving the demand for modern, cost-effective devices for their detection and monitoring. These gases pose a significant threat to human health, making accurate and continuous monitoring of their concentrations essential. Gas sensors, with their relatively low cost and ease of use, remain ideal for environmental monitoring, though their selectivity and durability still require further optimization to meet the needs of contemporary industries. In medicine, gas sensors are used for diagnosing diseases based on breath analysis. For instance, the detection of biomarkers like ammonia serves as an indicator of liver and kidney diseases, opening new possibilities for rapid, non-invasive diagnostics. This method could complement traditional blood and urine tests, providing a convenient and safe tool for health screening and monitoring [226]. Moreover, chemoresistive sensors show potential for food quality control, as they can detect volatile compounds that indicate spoilage, enabling automated monitoring systems that can be integrated into household devices [227].

Research into new synthesis methods and the effects of dopants on the gas-sensing properties of materials remains key to the development of multisensor systems. Despite the promising sensitivity of heterojunctions to NO_2_ under high humidity, the precise mechanisms of their operation are not yet fully understood. One of the main challenges for sensors in breath analysis is the high relative humidity, which can reach nearly 100%, significantly altering gas-sensing mechanisms. Under low humidity, sensors typically operate based on oxygen adsorption, but at high humidity, the mechanism shifts to physical water adsorption through the Grotthuss mechanism. This complicates the development of sensors that can effectively function under conditions similar to exhaled breath.

On the other hand, reducing energy consumption remains another critical challenge, as the high operating temperatures of metal oxide semiconductor sensors (100–450 °C) [228] present significant issues for practical use. These sensors require high temperatures, leading to reduced long-term stability, increased production costs, and higher energy consumption. These limitations make them less suitable for portable devices, where low power consumption and compact size are crucial. Recent studies have proposed various solutions, including the use of low-power LEDs [229], functionalization with noble metals [230], hybrid materials [231], self-heating modes, and integration with MEMS platforms [232]. However, despite these advancements, doping SMOx materials with transition metals and rare earth elements remains a promising direction for further improving sensor performance. This allows for the optimization of electrical and surface properties, thereby enhancing gas sensitivity and device stability. In advanced applications, semiconductors doped with rare-earth oxides are recognized as among the most suitable materials due to their enhanced selectivity and durability [233]. For example, studies have shown that doping metal oxides with praseodymium and europium improves the humidity resistance of sensors by neutralizing the effects of hydroxyl groups and restoring sensitivity to NO_2_ [234]. This approach could be the foundation for developing sensors that are independent of humidity levels.

At the same time, unresolved issues of sensor reproducibility, scalability, and sensitivity remain. Although commercial electrochemical sensors for NO_2_ detection are already available, they still struggle to reach ppb-level sensitivity [235]. Meanwhile, 1D carbon materials exhibit limited sensitivity to specific gases without surface modification. Functionalization or doping of materials such as SMOx and transition metal dichalcogenides can enhance their gas-sensing properties and address issues of durability and aggregation in carbon nanotubes and nanofibers. In this context, the electrospinning method can be a crucial tool. Electrospinning allows for the creation of nanofibers with a high surface area and unique morphology, which can significantly improve the sensitivity and selectivity of sensors by increasing the number of active centers for gas interaction.

However, sensor selectivity toward various gases remains a major challenge. Modern materials often show high sensitivity to one gas but low sensitivity to others. Detecting gases with similar physicochemical properties is even more difficult. Furthermore, the need to develop individual sensors for each specific gas makes the process labor-intensive and expensive, demanding further innovation [236].

In a recent study, researchers demonstrated the potential of using heterostructures based on 2D metal sulfides and oxides to enhance gas sensitivity due to the unique electronic properties at the interface. They showed that heterostructures of SnS and SnS_2_-SnO_2_ in the form of vertically oriented 1D nanostructures exhibited improved sensitivity to NO_2_ at high humidity (90% RH), with a theoretical detection limit of 1.67 ppt [237]. The moisture resistance of tin sulfides maintained active centers for gas interaction, ensuring high sensitivity.

The future of gas sensors lies in their integration with other sensing technologies, which will allow for increased sensitivity and the creation of more complex gas monitoring systems. The electrospinning method represents a promising direction, enabling the creation of new structures with unique properties, enhancing sensor sensitivity and selectivity. Success in gas sensor development will require an interdisciplinary approach, bringing together expertise in materials science, chemistry, and engineering. Through collaboration, it will be possible to integrate new materials and synthesis methods, leading to the creation of more efficient sensor systems for a wide range of applications [238].

## 6. Conclusions

This review article highlights the critical role of gas sensors, particularly those based on electrospun nanofibers, in ensuring safety and air quality control across various fields—from industry to medicine. The electrospinning technique enables the formation of nanofibers with unique morphologies characterized by high surface area and porous structures. These features significantly enhance key gas-sensing properties of sensors, such as sensitivity, selectivity, and response time.

Despite substantial advancements in nanomaterials and technologies, nanofiber-based sensors face significant challenges, including limited selectivity for specific gases and difficulties in scaling up for mass production. Promising research directions include optimizing fiber morphology, incorporating functional materials, and developing advanced post-annealing techniques such as surface modification or doping with active elements. These approaches are expected to improve the selectivity and stability of sensors, unlocking new possibilities for their effective application in real-world conditions.

Thus, focusing on the precise tuning of electrospun fiber morphology and its impact on gas-sensing properties is a key aspect of advancing gas monitoring technologies. It is anticipated that a detailed analysis of new materials and methods will overcome existing limitations, significantly enhancing the performance of gas sensors and broadening their application across various industries.

## Figures and Tables

**Figure 1 sensors-24-06797-f001:**
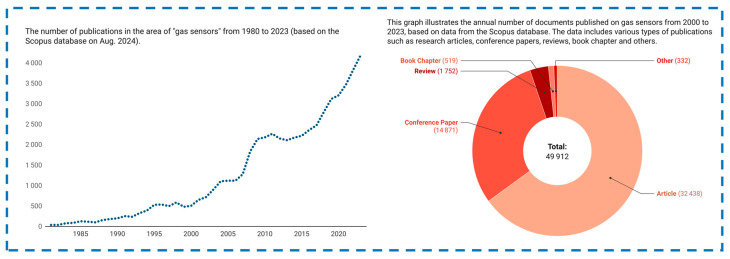
The number of publications on gas sensors from 1980 to 2023, based on data from the Scopus database.

**Figure 2 sensors-24-06797-f002:**
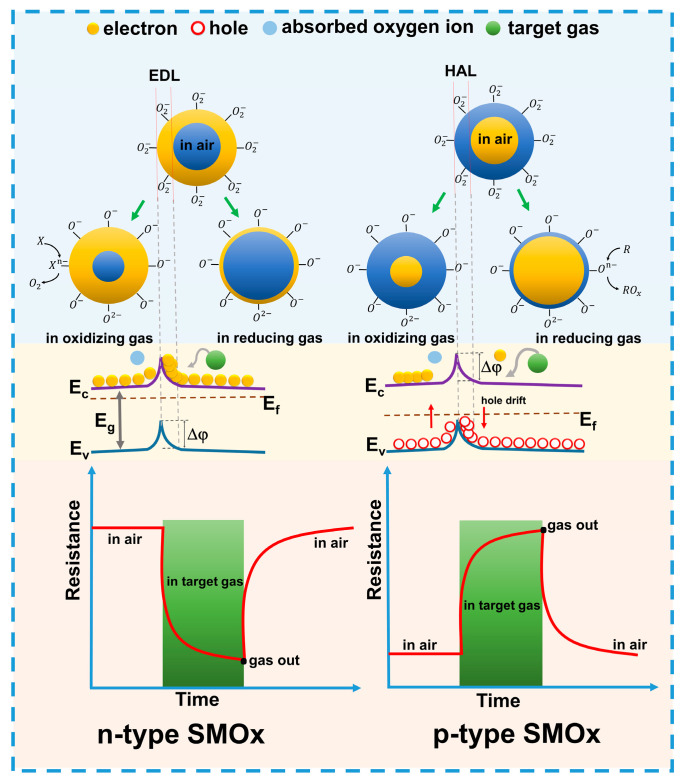
Schematic illustration of the mechanism of interaction of gases with electrons and holes in semiconductor metal oxides.

**Figure 3 sensors-24-06797-f003:**
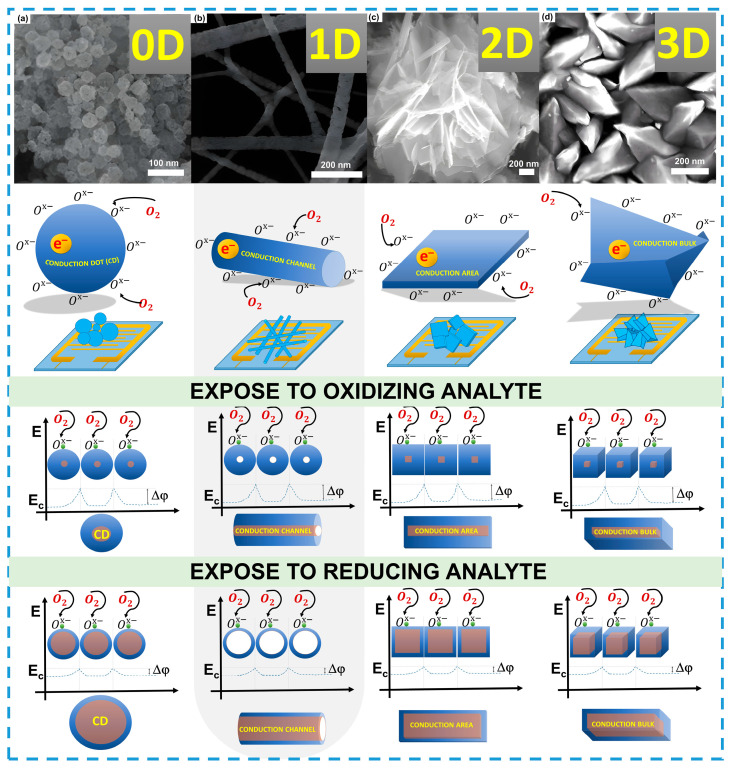
Mechanisms of electronic transport in nanomaterials of different dimensionalities. zero-dimensional nanospheres (**a**), reprinted with permission from Ref. [108]. One-dimensional nanofibers (**b**), reprinted with permission from Ref. [111]. Two-dimensional nanoflowers (**c**), reprinted with permission from Ref. [112]. Three-dimensional nanostructures (**d**), reprinted with permission from Ref. [109] during interaction with analytes.

**Figure 4 sensors-24-06797-f004:**
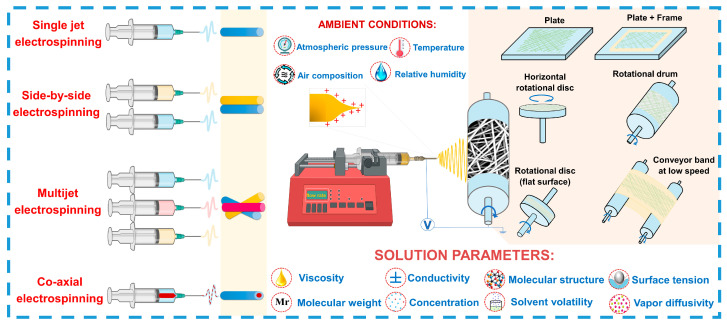
Schematic illustration of the electrospinning process.

**Figure 5 sensors-24-06797-f005:**
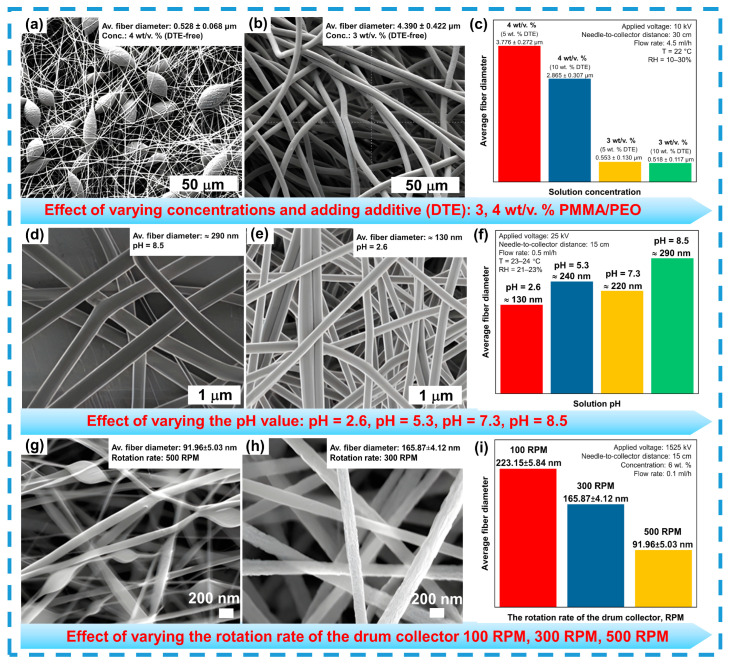
Effect of different conditions on nanofiber morphology and diameter: solution concentration (**a**–**c**), reprinted with permission from Ref. [124]. Solution’s pH (**d**–**f**), reprinted with permission from Ref. [131]. Collector speed (**g**–**i**), reprinted with permission from Ref. [132].

**Figure 6 sensors-24-06797-f006:**
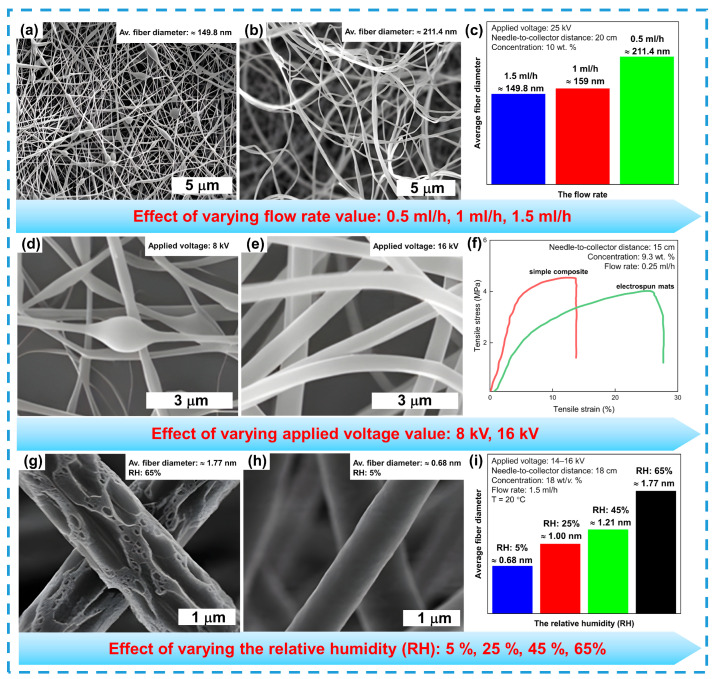
Effect of different conditions on nanofiber morphology: flow rate (**a**–**c**), reprinted with permission from Ref. [136]. Applied voltage (**d**–**f**), reprinted with permission from Ref. [138]. Relative humidity (**g**–**i**), reprinted with permission from Ref. [139].

**Figure 7 sensors-24-06797-f007:**
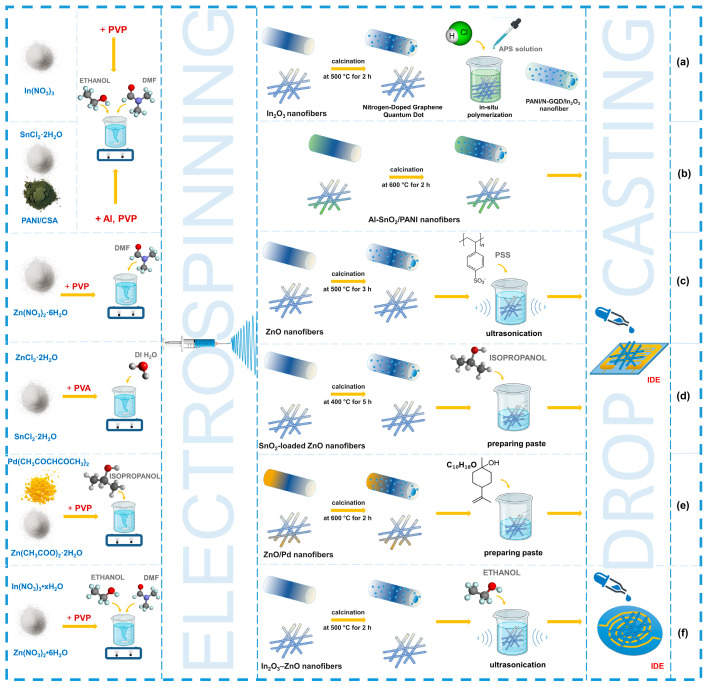
Fabrication of integrated electrodes based on electrospun fibers using the drop-casting method: (**a**) adapted with permission from Ref. [147]. (**b**) adapted with permission from Ref. [148]. (**c**) adapted with permission from Ref. [149]. (**d**) adapted with permission from Ref. [150]. (**e**) adapted with permission from Ref. [151]. (**f**) adapted with permission from Ref. [152].

**Figure 8 sensors-24-06797-f008:**
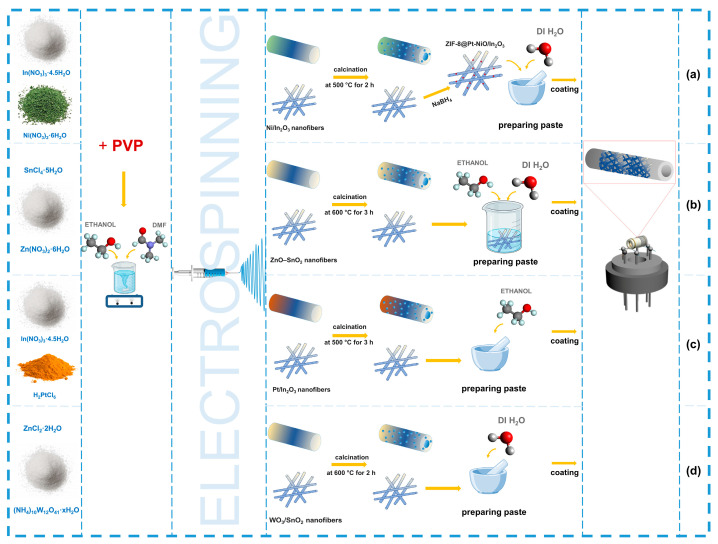
Fabrication of tubular electrodes based on electrospun fibers using coatings: (**a**) adapted with permission from Ref. [155]. (**b**) adapted with permission from Ref. [156]. (**c**) adapted with permission from Ref. [157]. (**d**) adapted with permission from Ref. [158].

**Figure 9 sensors-24-06797-f009:**
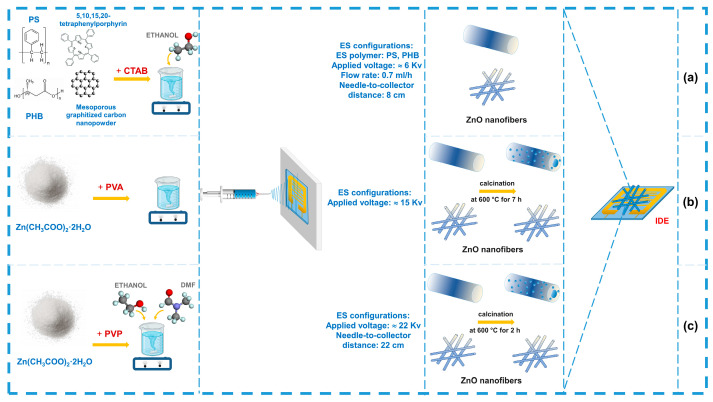
Fabrication of electrodes on the collector: (**a**) adapted with permission from Ref. [159]. (**b**) adapted with permission from Ref. [160]. (**c**) adapted with permission from Ref. [161].

**Figure 10 sensors-24-06797-f010:**
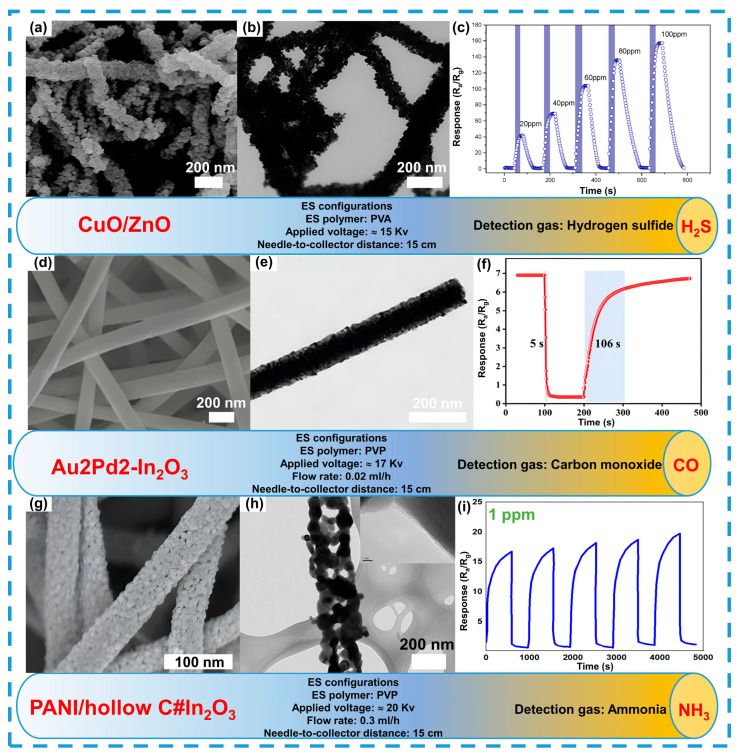
Morphology and structure of composite fibers as an example of reductive gas detection. Detection to H_2_S (**a**–**c**), reprinted with permission from Ref. [170]. CO (**d**–**f**), reprinted with permission from Ref. [171]. NH_3_ (**g**–**i**), reprinted with permission from Ref. [172].

**Figure 11 sensors-24-06797-f011:**
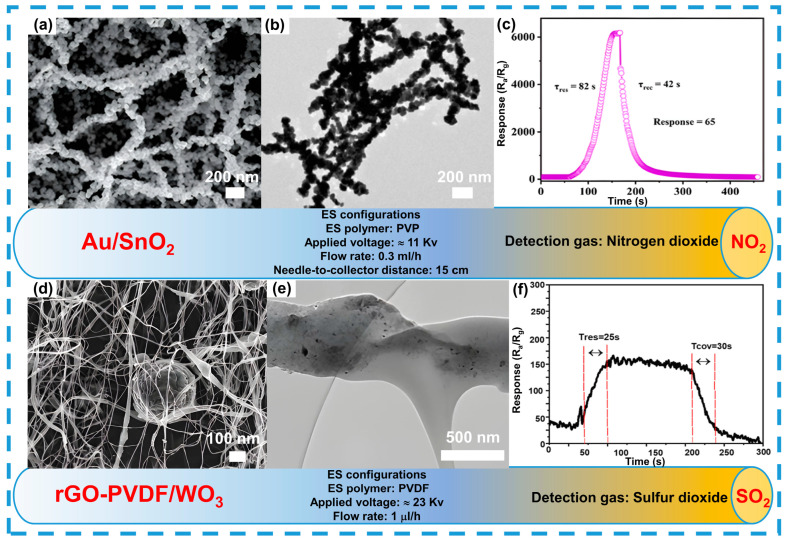
Morphology and structure of composite fibers as an example of oxidizing gas detection. Detection to NO_2_ (**a**–**c**), reprinted with permission from Ref. [192]. SO_2_ (**d**–**f**), reprinted with permission from Ref. [193].

**Figure 12 sensors-24-06797-f012:**
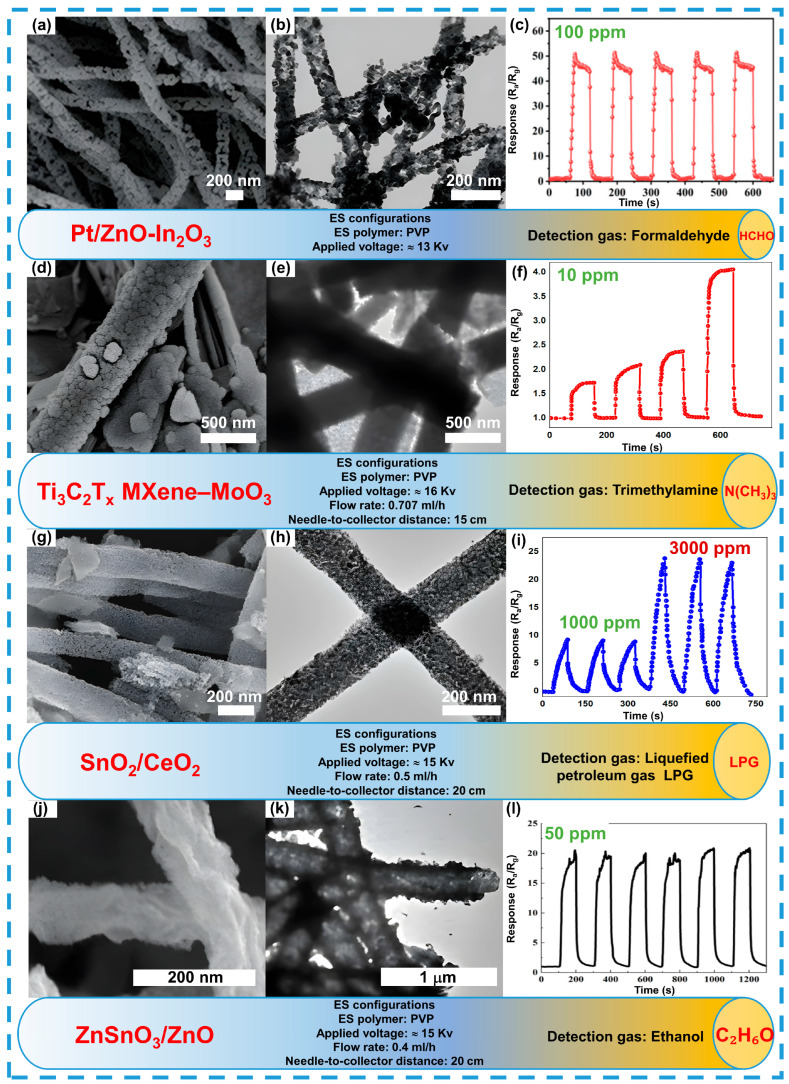
Morphology and structure of composite fibers as an example of VOCS detection. Detection to HCHO (**a**–**c**), reprinted with permission from Ref. [208]. N(CH_3_)_3_ (**d**–**f**), reprinted with permission from Ref. [209]. LPG (**g**–**i**), reprinted with permission from Ref. [210]. Ethanol (**j**–**l**), reprinted with permission from Ref. [211].

**Table 1 sensors-24-06797-t001:** Main characteristics of gas sensors.

Characteristics	Operating Principle	Major Limitations	Potential Solutions
Sensitivity	- Change of electrical conductivity of the conductor layer during gas adsorption - Change in mass of the sensing element (for piezoelectric sensors) - Optical absorption (for optical sensors)	- Low signal-to-noise ratio for trace gases - Interference from other gases - Influence of temperature and humidity	- Nanostructured materials (e.g., SnO_2_ ZnO nanoparticles [59,60]) - Noble metal alloying (e.g., Pd, Au) [61,62] - Composite materials (e.g., ZnO@CO_3_O_4_ [63]) - Optimization of the operating temperature [64,65]
Selectivity	- Specific chemical reactions on the sensor surface - Size matching between pores and gas molecules - Differences in adsorption/desorption kinetics for different gases	- Cross-sensitivity to similar gases - Difficulty in distinguishing between homologous gases (e.g., CO and H_2_)	- Functionalisation by specific catalysts [66] - Selective filter membranes (e.g., ZIF-8 [67]) - Composite materials with opposite responses (e.g., SnO_2_-NiO for VOCs [68])
Response time	- Adsorption/desorption rate of gas on the sensor surface - Gas diffusion rate in porous material - Rate of chemical reaction on the surface	- Slow adsorption/desorption kinetics - Diffusion limitations in porous materials	- Nanostructured materials with high specific surface area [69] - Optimization of operating temperature [70] - Thin sensing layers [71] - Catalytic additives for reaction acceleration [72]
Stability	- Preservation of physicochemical properties of the sensing layer in time - Reversibility of adsorption/desorption processes - Stability to environmental changes	- Drift of readings over time - Poisoning by interfering gases - Degradation at high temperatures	- Protective coatings (e.g., PMMA membranes) [73] - Stable nanostructures (e.g., 1D nanowires [74,75]) - Periodic recalibration [76,77]

**Table 2 sensors-24-06797-t002:** Comparison of electrospun-fiber-based sensors for detecting reducing gases.

Materials	ES Parameters			Target Gas	Response Time	Recovery Time	Operating T°C	Selectivity	Detection Range, ppm	Sensitive Concentration, ppm	Ref.
	Flow Rate, mL/h	Voltage, kv	Needle-to- Collector Distance, cm								
SnO_2_-loaded ZnO	0.01	15	20	H_2_	–	–	300	H_2_, CO, NO_2_	0.05–5	5	[150]
ZnO	–	22	22	H_2_	–	–	210–330	H_2_	20–100	100	[160]
PANI/PEO/ZnO	1.3	25	20	NH_3_	245	153	RT	H_2_, H_2_S	250	250	[176]
p-NiO-loaded n-ZnO	0.02	15	20	H_2_	–	–	200	H_2_, H_2_S, CO, C_6_H_6_	0.1–10	10	[177]
PVA/PEDOT:PSS	10 m s^−1^	20	15	NH_3_	10	–	RT	NH_3_	50	50	[178]
ZnO	0.56	12	15	H_2_S	14	49	180	H_2_S, VOCs, NH_3_	50	50	[179]
PPy-PAN	0.8	10	20	NH_3_	1	60	RT	VOCs	250–2000	2000	[180]
NiO/SnO_2_	1	12	15	H_2_	12	5	195	H_2_	25–100	25	[181]
CuO-SnO_2_	0.4	8	12	H_2_S	284	539	150	NO, CO, CH_4_, SO_2_, C_2_H_5_OH	1–10	10	[182]
CFs@NiNPs–PtNPs	1	25	21	H_2_	24	89	RT	H2, NH_3_	10,000–40,000	1000	[183]
SnO_2_	–	17	–	H_2_S	15	230	350	H_2_S, CO, H_2_, SO_2_, NH_3_	0.1–1	1	[184]
Cu/CuO@ZnO	1.2	20	15	CO	–	–	400	–	–	100	[185]
Co_3_O_4_	0.707	20	10	CO	14	36	100	CO, NO_2_, H_2_, CH_4_, NH_3_	5–40	5	[186]

**Table 3 sensors-24-06797-t003:** Comparison of electrospun-fiber-based sensors for detecting oxidizing gases.

Materials	ES Parameters			Target Gas	Response Time	Recovery Time	Operating T°C	Selectivity	Detection Range, ppm	Sensitive Concentration, ppm	Ref.
	Flow Rate, mL/h	Voltage, kv	Needle-to- Collector Distance, cm								
ZnO/Bi_2_O_3_ ZnO/In_2_O_3_	0.8–1	20	18	NO_2_	5–7	–	200	NO_2_	0.5–3	0.5	[194]
SFRGO	0.5	20	15	NO_2_	–	–	RT	NO_2_, VOCs	0.01–20	20	[195]
Au-PANI/ZnO	0.5	16	14	NO_2_	–	–	300	H_2_, NO_2_, CO, NH_3_	10–50	50	[196]
WO_3_	0.5	14.5	15	NO_2_	11	26	200	NO_2_, VOCs	0.2–50	1	[197]
PdOx@SnO_x_	0.3	1.2	–	NO_2_	–	–	325	NO_2_, CO, NH_3_	0.0625–0.25	0.25	[198]
SnO_2_/ZnO	–	20	20	NO_2_	126	–	RT	NO_2_, SO_2_, CO, NH_3_, VOCs	0.1–2	0.5	[199]
PANI/g-C_3_H_8_/PVDF	0.5	15	17	NO_2_	–	–	RT	NO_2_, NH_3,_ VOCs	8–108	108	[200]
WO_3_	0.06	20	15	NO_2_	15 min	0.8 min	150	NO_2_, H_2_, CO	2–25	25	[201]
rGO-PVDF/WO_3_	0.001	23	10	SO_2_	25	30	200	SO_2_, NH_3_, CO_2_, VOCs	5–80	80	[193]
MoS_2_/SnO_2_	–	17	13	SO_2_	–	–	150	SO_2_, CO, H_2_, NH_3_	1–10	10	[202]
Zr-MOF	–	–	–	SO_2_	185	–	RT	SO_2_	1–150	50	[203]

**Table 4 sensors-24-06797-t004:** Comparison of electrospun-fiber-based sensors for detecting VOCs.

Materials	ES Parameters			Target Gas	Response Time	Recovery Time	Operating T°C	Selectivity	Detection Range, ppm	Sensitive Concentration, ppm	Ref.
	Flow Rate, mL/h	Voltage, kv	Needle-to- Collector Distance, cm								
Pt-SnO_2_	0.003	15	15	Acetone	13	24	150	VOCs	0.1–20	2	[213]
Rh-SnO_2_	0.3	13	13	Acetone	2	64	200	VOCs	90–200	50	[110]
SnO_2_/ZnO	–	20	15	Acetone	12 s	27 s	350	VOCs	1–100	5	[214]
Pt-SnO_2_	0.03	15	20	C_7_H_8_	–	–	300	C_6_H_6_, C_7_H_8_, CO	1–10	10	[215]
ZnO	–	18	20	Acetone	40	30	260	VOCs		50	[216]
Co–CeO_2_@SnO_2_	0.3	16	10	C_5_H_8_	5 s	514 s	350	VOCs	0.1–5	5	[217]
MoO_3_-WO_3_	1	20	15	Acetone	–	–	375	VOCs, NH_3_·H_2_O	20–1000	100	[218]
PANI/P_3_TI/PMMA	0.6	20	10	n-Butanol	10	–	RT	n-Butanol, CB, DMF, n-Propanol, Toluene	100–2000	100	[219]
CuO	0.3	–	12	VOCs	–	–	RT	H_2_, Ethanol, LPG	50–350	350	[220]
Pd-CeO_2_	0.5	15	15	Methanol	1	5	200	H_2_, NH_3_, CO, VOCs	5–2000	100	[221]
Au-SnO_2_	0.008	11	5	Tetrahydrocannabinol	–	–	350	Tetrahydrocannabinol, Methanol	200–1000	1000	[222]
Co_3_O_4_	0.016	7	7	Methanol	15	26	350	VOCs	21–2094	4–2094	[223]
EPS/rGO	1	15	20	Ethanol	110	20	RT	Ethanol, acetone, toluene	10–80	10	[224]
Pd@Co_3_O_4_-ZnO	–	10	10	Ethanol	6	12	240	Ethanol, acetone, isopropanol	1–2000	200	[225]

## Data Availability

Data are contained within the article.

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
