# Peer review of "Chemoresistive Gas Sensors Based on Electrospun 1D Nanostructures: Synergizing Morphology and Performance Optimization"

_sensors, 2024, doi:10.3390/s24216797_

Round 1

Reviewer 1 Report

Comments and Suggestions for Authors

This article reviewed the chemoresistive gas sensors based on electrospun 1D nanostructures, focusing on the influence of material morphology and composition on sensor properties. The review is comprehensive and presents the current status of research on chemoresistive gas sensors using one-dimensional nanostructured materials. There are some minor writing issues that need improvement

1. In Line 129, the chemical formula oxygen ions for the first and third are the same, please correct.

2. Reference formatting should be standardized, e.g., some chemical formulas are not marked with subscripts.

Author Response

Dear Reviewer,

Thank you for your constructive feedback on our article. We appreciate your recognition of the comprehensive nature of our review regarding chemoresistive gas sensors based on electrospun 1D nanostructures.

We have made the necessary corrections you pointed out:

Comment 1: In Line 129, the chemical formula oxygen ions for the first and third are the same, please correct..

Response 1: The chemical formula for oxygen ions has been corrected in Line 129.

Comment 2: Reference formatting should be standardized, e.g., some chemical formulas are not marked with subscripts.

Response 2: We have standardized the reference formatting throughout the manuscript to ensure that all chemical formulas are appropriately marked with subscripts.

Your suggestions are invaluable for enhancing the quality of our work. 

Best regards,

Gaukhar S.

Reviewer 2 Report

Comments and Suggestions for Authors

The article reviews metal oxide nanofiber gas sensors obtained by electrospinning. It outlines the main benefits of creating gas sensors using one-dimensional nanostructures and provides an overview of the principles, characteristics, and materials used in semiconductor gas sensors.

The paper discusses the challenges of electrode formation and the use of electrospinning to create gas sensors. It also provides an overview of recent developments in this field.

1. In Fig. 2, for p- and n-type semiconductors, the Fermi level is the same. The authors should correct the error.

2. In table 1 (potential solutions for selectivity), the authors should replace the word “catalytic-lysers”.

3. In subsection 4.2, examples of materials obtained by electrospinning for the detection of oxidizing gases are considered. Figure 10 and Table 3 show CO, which is a typical reducing gas, as an example of an oxidizing gas. The authors should eliminate this inaccuracy and transfer this example to reducing gases (subsection 4.1).

4. The authors have allocated sensors for detecting volatile organic compounds in a separate subsection 4.3, however, a representative of this class, ethanol, is included in the subsection of reducing gases (4.1). The authors should eliminate this inaccuracy and move this example to the appropriate part.

Author Response

Dear Reviewer,

We would like to express our sincere gratitude for your valuable feedback on our manuscript.

We appreciate the opportunity to address your comments and are pleased to inform you that the following corrections have been made:

Comment 1: In Fig. 2, for p- and n-type semiconductors, the Fermi level is the same. The authors should correct the error.

Response 1: The error (in Fig. 2) regarding the Fermi level has been corrected to accurately represent the differences between p-type and n-type semiconductors.

Comment 2:  In table 1 (potential solutions for selectivity), the authors should replace the word “catalytic-lysers”.

Response 2: The term (in Table 1) “catalytic-lysers” has been replaced with “catalysts” to ensure the use of appropriate terminology.

Comment 3, 4: 3. In subsection 4.2, examples of materials obtained by electrospinning for the detection of oxidizing gases are considered. Figure 10 and Table 3 show CO, which is a typical reducing gas, as an example of an oxidizing gas. The authors should eliminate this inaccuracy and transfer this example to reducing gases (subsection 4.1).

  1. The authors have allocated sensors for detecting volatile organic compounds in a separate subsection 4.3, however, a representative of this class, ethanol, is included in the subsection of reducing gases (4.1). The authors should eliminate this inaccuracy and move this example to the appropriate part.

Response 3, 4: We have made the necessary adjustments as per your suggestions. The characteristics of ethanol have been moved to Subsection 4.3, while those of CO have been transferred to Subsection 4.1. The relevant tables have been updated accordingly.

Your careful review has significantly contributed to the accuracy and clarity of our manuscript. Thank you once again for your insightful input.

Best regards,

Gaukhar S.

Reviewer 3 Report

Comments and Suggestions for Authors

The review article may be interesting for the scientistы working in the ield of chemoresistive sensors. From my point of view large work has been done. But there are some comments:

1)The composition of 1 D materials is similar to that of powders and films, which are also used to create gas sensors. The author needs to select a section reflecting the features of 1D materials in comparison with others

2)The authors have highlighted sections describing sensors for reducing gases, oxidants and vapors of organic substances. In many cases, we are talking about the same materials. For a clearer picture, it is necessary to clarify the mechanisms of gas sensitivity of 1 D materials in comparison with other materials of the same composition

Author Response

Dear Reviewer,

We sincerely appreciate your thoughtful feedback on our manuscript. We have addressed your comments as follows:

Comment 1: The composition of 1 D materials is similar to that of powders and films, which are also used to create gas sensors. The author needs to select a section reflecting the features of 1D materials in comparison with others

Response 1: To provide a more detailed understanding of the gas sensor mechanisms, we have added a new section (2.4) that focuses on the structure of 0D, 1D, 2D, and 3D materials. This section highlights the key differences in electron transfer processes in 1D nanostructures compared to other types of materials (as illustrated in Figure 3).

Comment 2: The authors have highlighted sections describing sensors for reducing gases, oxidants and vapors of organic substances. In many cases, we are talking about the same materials. For a clearer picture, it is necessary to clarify the mechanisms of gas sensitivity of 1 D materials in comparison with other materials of the same composition

Response 2: We have also expanded on the mechanisms of gas sensitivity for 1D materials, comparing them with other materials of the same composition. This addition is intended to offer a clearer perspective on the unique properties and advantages of 1D nanostructures in gas sensing applications.

Thank you once again for your insightful comments, which have significantly enhanced the clarity and depth of our manuscript.

Best regards,

Gaukhar S.

Reviewer 4 Report

Comments and Suggestions for Authors

Dear authors,

I begin by saying that it was a pleasure to read your review about chemoresistive sensors obtained by electrospinning technique.

The manuscript is well structured and presents a detailed analysis of the research in this field.

In my opinion a small comment to chapter 2 (Basics of chemometrics gas sensors), page 4, the paragraph between lines 158-166 is recommended. The explanations presented by the authors are correct, but it could be add that the interactions between the sensing substrate and the gas molecules whose concentration is measured are due to a combination of mechanisms that differ depending on the substrate and the measured gas. Besides that, Pearson's HSAB theory can be used as a way to explain the interaction of the substrate with the measured gases, but also as a tool for predicting the architecture of the chemoresistive sensor depending on the measured gas.

Congratulations

Author Response

 Dear Reviewer,

Thank you for your valuable comments regarding chapter 2 of the manuscript.  

Comment 1: In my opinion a small comment to chapter 2 (Basics of chemometrics gas sensors), page 4, the paragraph between lines 158-166 is recommended. The explanations presented by the authors are correct, but it could be add that the interactions between the sensing substrate and the gas molecules whose concentration is measured are due to a combination of mechanisms that differ depending on the substrate and the measured gas. Besides that, Pearson's HSAB theory can be used as a way to explain the interaction of the substrate with the measured gases, but also as a tool for predicting the architecture of the chemoresistive sensor depending on the measured gas.

Response 1: We appreciate your insights on the role of acid-base centers in gas detection. In response to your suggestion, we have added a discussion on the various mechanisms of interaction between the sensing substrate and gas molecules (lines 169-175). Additionally, we have applied Pearson's Hard and Soft Acid-Base (HSAB) theory to further explain these interactions and to guide the optimization of gas sensor architecture. We believe these additions have enhanced the clarity and depth of this section, and we are grateful for your input.

Thank you once again for your constructive feedback.

Best regards,

Gaukhar S.

Round 2

Reviewer 3 Report

Comments and Suggestions for Authors

i am thank the authors for reviewing the article according to the comments

now i recommend the article for publication